# The Typology of Dubrovnik Summer Residences as a Spatial Planning Tool for Developing the Coexistence of Privacy and Sociality: A Case Study of the Gruž Area

**Marijana Jurić, Mia Jurić and Krunoslav Šmit** *

Faculty of Architecture, University of Zagreb, Fra Andrije Kačića Miošića 26, 10000 Zagreb, Croatia; marijana.juric@arhitekt.hr (M.J.); mia.juric@arhitekt.hr (M.J.)
* Correspondence: krunoslav.smit@arhitekt.hr; Tel.: +385-98-9066-641

**Abstract:** The architecture of summer residences in the Dubrovnik region from the 15th and 16th centuries represents elements of cultural and historical heritage that both enhance the landscape and bear witness to the rich legacy rooted in the native Mediterranean ambience. By learning about the specific spatial characteristics that define Dubrovnik's summer residences, this research aimed to comprehend their urban and architectural essence and determine the possibilities of using the typology of these residences as a tool for planning the balanced development of both the private and societal aspects of the city. This study identified indicators of the spatial parameters of existing historic Dubrovnik summer residences in Gruž and then analyzed them according to types of spatial planning conditions to guide the construction and development of building plots. The research was conducted using a model representation of the spatial indicators of the summer residences. The analysis of the model data revealed the characteristics of the typology of Dubrovnik summer residences, highlighting the urban and architectural features of the plots, houses, and gardens, the use of technological innovations, and the coexistence of privacy and sociality. The recognized specificities led to the conclusion that the typology of Dubrovnik summer residences can serve as an exceptionally valuable spatial planning tool.

**Keywords:** typology; Dubrovnik summer residences from the 15th and 16th centuries; Gruž; spatial indicators; plot; house; garden; technological innovations; privacy and sociality; spatial planning tool

## 1. Introduction

The spatial development of the City of Dubrovnik has been shaped by its rich history, proximity to the sea, and connectedness with the surrounding areas. This development represents an interesting story about balancing urban expansion and preserving the coastline and the islands, as well as thoughtful hinterland development. In Dubrovnik, the area of Gruž Bay is organized as the main city harbor and a trading center. However, it is also a city district where a large number of historic summer residences are located.

The architecture of summer residences in the Dubrovnik region represents elements of cultural and historical heritage that both enhance the landscape and bear witness to the rich legacy rooted in the native Mediterranean ambience. These magnificent complexes, characteristic of the Dubrovnik coastal region, not only served as destinations for a summer getaway from the city's bustle but also reflected the social status and aesthetic preferences of their owners (Figure 1).

By familiarizing ourselves with the specific spatial characteristics that define Dubrovnik's summer residences, this research aimed to comprehend their urban and architectural essence and determine the possibilities of using the typology of these residences as a tool for planning the balanced development of both the private and societal aspects of the city, in accordance with the paradigms of the New European Bauhaus [1,2].

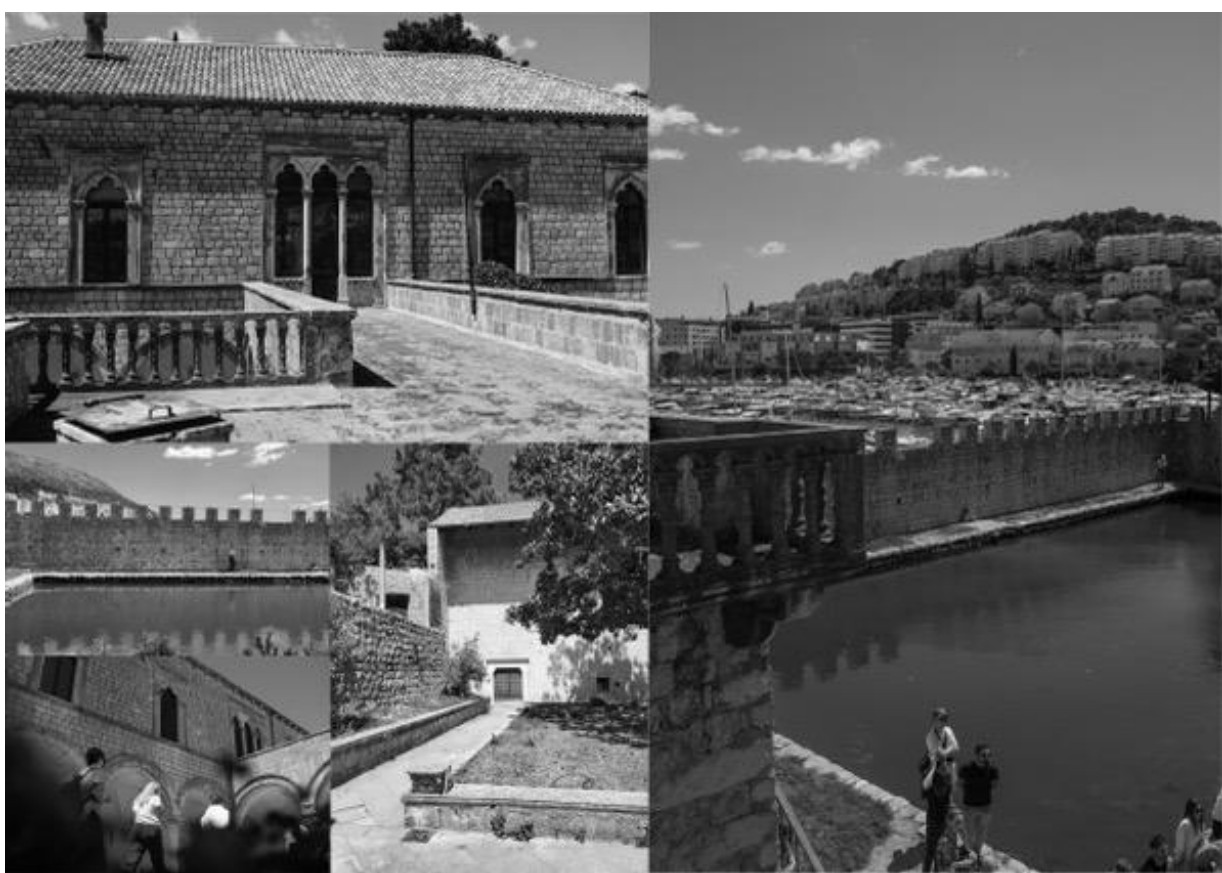

**Figure 1.** The summer residence of Petar Sorkočević—a collage of experiential environments, based on photographs by Bruno Žganjer Šram and Marijana Jurić. Figure created by the author.

Historic Dubrovnik summer residences (DSRs) stand out for their harmonious integration with their natural surroundings, employing local materials such as stone and wood to achieve an authentic aesthetic [3]. Spacious terraces, arcades, and water features allowed for excellent air circulation and ensured a comfortable stay during the warm summer months. As a result, these summer residences have become known for their ability to provide intimacy and comfort within their walls, which were also open to their surrounding environments [4–7] (Figure 2).

In 1440, Filip de Diversis described Gruž as a place "(...) where the view opens to the most sheltered and magnificent harbor, curved like an arch and adorned all around with numerous fertile vineyards, majestic palaces, and beautiful gardens". In the same section, it was stated that there are attractive vineyards and private residences with gardens on both sides of the part of the city known as Rijeka Dubrovačka. The choice of words is deliberate: in Gruž, the text refers to "grand palaces and delightful gardens", whereas in Rijeka Dubrovačka, it only mentions "private houses with gardens". Archival sources indicate that summer residences in Gruž Bay were already present in the 14th century, and by the mid-15th century, the bay had become an exclusive area for Dubrovnik's summer retreats [5].

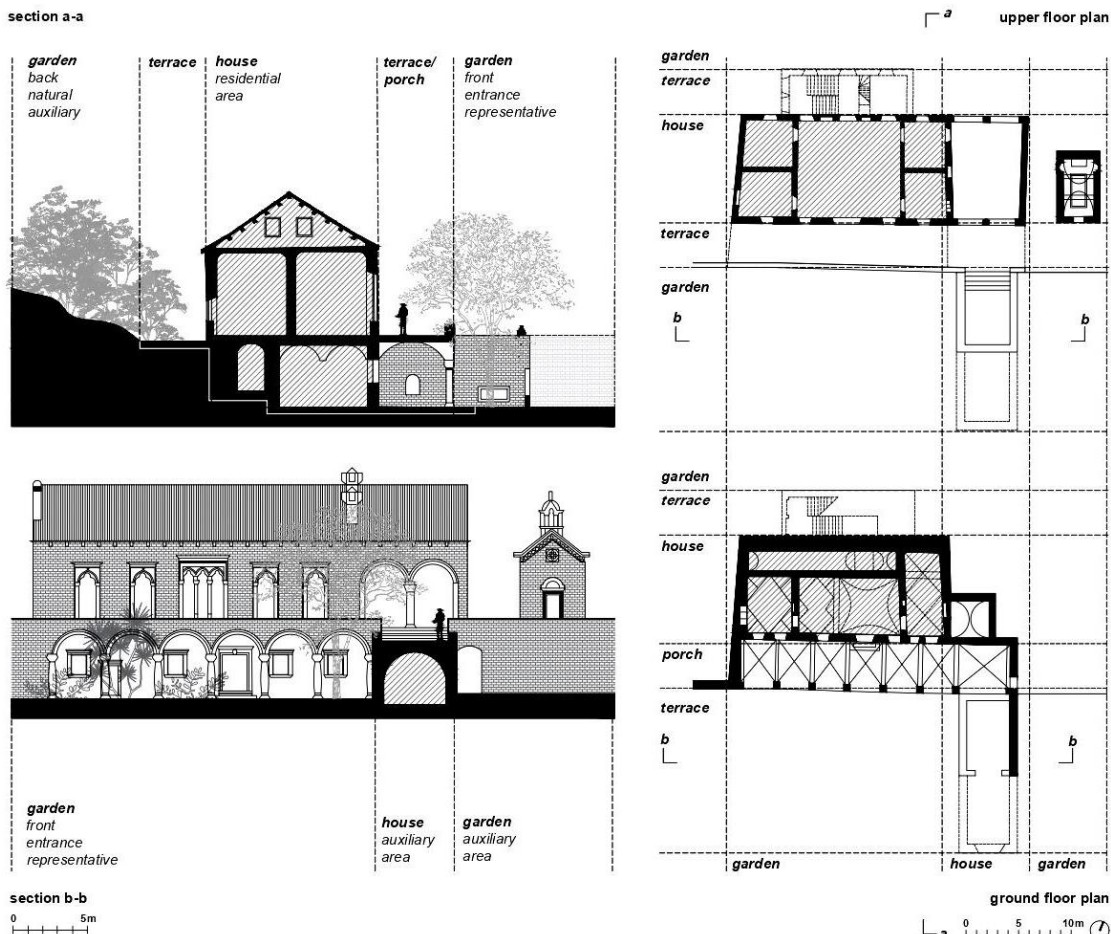

**Figure 2.** The summer residence of Miho Bunić—floor plans and sections, based on research by Nada Grujić [8]. Figure created by the authors.

The Gruž coastline has undergone numerous urban transformations, from historical industrial developments to contemporary tourism-related changes [9]. This has included the development of a nautical marina and a cruise port, along with an increase in accompanying service and accommodation capacities, as well as planned spatial development [10]. One of the problems that arise during the construction of new buildings on the coast stems from the phenomenon of mass tourism and its negative impact on the local population. This has been corroborated by studies that have explored the coastal areas with a focus on tourism issues, traffic, and the disruption of the coastline's urban fabric [11]. Therefore, one can raise the question of whether new urban complexes can take on the task of mitigating the negative impacts of tourism and building upon the spatial characteristics of Dubrovnik's summer residences while promoting the coexistence of privacy and sociality [12]. This topic becomes important as efforts are made to preserve the authentic urban environments and specific spatial features when undertaking new construction projects along the coastal areas of the city. We are witnessing more and more changes in the existing environments where new buildings are emerging and occupying previously undeveloped urban areas, creating heterogeneous spaces with different environmental, architectural, and urban expressions [10].

The importance of researching this issue is further confirmed in the Management Plan for the Protected Heritage Complex of the City of Dubrovnik, which was developed with the aim of striking a balance between the protection and restoration of the architectural heritage and the economic development, functionality, and vitality of the city, and not just for the preservation of the historical core and historical buildings. In doing so, it is emphasized that approaches to protect various aspects (heritage-based, environmental, eco-

nomic, and sociocultural) should be mutually complementary to preserve the authenticity of Dubrovnik's historical layers [13]. The most significant challenges in management can be found in thematic areas: planning, protection and preservation, sustainable development, tourism, transportation, and risks [14]. There is a distinct need for collaborative urban and environmental development planning with an equal measure of problem assessment. It has been suggested that protection and preservation should find a balance between urban expansion and the preservation of the historical urban landscape, which is an important component of UNESCO cultural heritage. Proposed development solutions should achieve harmony between the new and the old, emphasizing all aspects of Dubrovnik's outstanding universal value. Emphasis has been placed on the fact that the sustainable development of Dubrovnik is imperative for preserving its authenticity and the quality of life of the local population. The suggestion is to focus on socially, economically, environmentally, and culturally enriching tourism that would play a significant role in local communities. It has been observed that sustainable traffic organization in Dubrovnik is a crucial need because the city is facing challenges of above-average traffic congestion, especially during summer crowding. The risks are recognized, as well as the need to ensure a healthy and safe environment to address climate change, which includes reducing pollution, protecting natural resources such as the sea and air, and preserving the city's parks and green areas.

Considering all this, it is necessary to identify the highest-quality and most unique forms of construction while determining the possibilities of their contemporary urban counterparts, as has been indicated in research concerning the impact of sociocultural factors on changes in traditional residential architecture [15]. Previous experiences have indicated the possibility of losing authenticity and cohesion in the urban fabric due to the lack of specifically designed guidelines for using space and construction. They have also highlighted the need for the establishment of improved planning tools that will encourage the development of more layered forms of sociality integrated into urban environments [13].

The aim of this paper is to determine the characteristics of historic DSRs expressed through contemporary spatial parameters for urban development planning. By understanding the past and adapting to the present, the goal is to shape complexes that will not only inherit the aesthetics and function of the original summer residences but also contribute to the preservation of their identity in creating recognizable forms of coexistence between privacy and sociality along the Dubrovnik coastline [16].

## 2. Materials and Methods

This study identified the distinctive indicators of the spatial parameters of the existing historic DSRs in Gruž (Figure 3) and then analyzed them according to types of spatial planning conditions to guide the construction and development of building plots [16]. A total of 18 historic summer residences were analyzed: Ghetaldi–Gondola–Solitudo, Pucić–Kosor, Petar Sorkočević, Gundulić–Zago, Bonda–Majstorović, Gradi–Vuić, Junije Bunić, Marin Bunić, Paladin Gundulić, Kaboga–Zec, Bobaljević–Pucić, Natali Sorkočević, Miho Bunić, and Stay. The analysis excluded the summer residences whose plots and gardens had been confiscated over time (Giorgi–Matijević and Bassegli Gozze) and those that had been converted into hotel and hospitality buildings (Zamagna–Kazbek and Pucić–Pitarević). The research was conducted using a model representation of the spatial indicators of the summer residences (Figure 4) based on data from cadastral maps and land registries [17], topographic maps, satellite imagery, and urban plans [10,18–20], a 3D city model [21], and the literature, as well as textual data sources [3–9,13].

The model provides plans for the spatial organization of DSR complexes on the Gruž coast in order to define their spatial composition, their floor layout disposition, the relationships between the built, residential, and auxiliary parts of the complexes, and the landscaped areas of the plots.

The spatial composition of the plots, having a prominent relationship with the sea and the coastline, was based on graphical data from satellite maps, which were cross-referenced and supplemented with information from topographic maps and urban plans.

The floor layout of the buildings on the plot was based on data from cadastral maps and representations from the 3D city model. The surface areas of the plots and the built portions of the complexes were extracted as numerical data from land registries. The surface areas of the landscaped parts of the plots were obtained by calculating the difference between the total plot surface area and the surface area of the built portions of a complex, with additional information derived from satellite maps. Literature and textual data sources were used to complement and verify the graphical elements of the model's representation, as well as to establish numerical data regarding the characteristics of the summer residences. Additionally, a field survey was conducted across the entire area of the researched summer residences to confirm all the data from the model's representation.

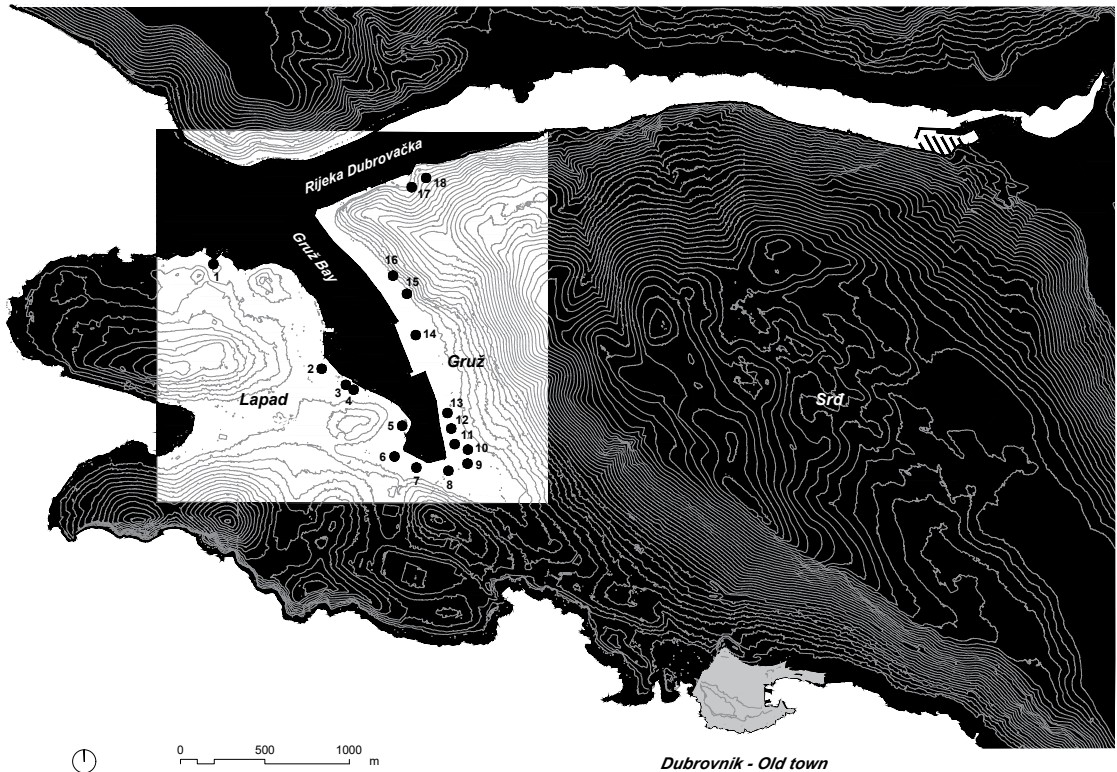

**Figure 3.** Spatial distribution of Dubrovnik summer residences in the Gruž area: 1, Ghetaldi–Gondola–Solitudo; 2, Pucić–Kosor; 3, Zamagna–Kazbek; 4, Pucić–Pitarević; 5, Petar Sorkočević; 6, Giorgi–Matijević; 7, Gundulić–Zago; 8, Bonda–Majstorović; 9, Gradi–Vuić; 10, Bassegli Gozze; 11, Junije Bunić; 12, Marin Bunić; 13, Paladin Gundulić; 14, Kaboga–Zec; 15, Bobaljević–Pucić; 16, Natali Sorkočević; 17, Miho Bunić; and 18, Stay. Figure created by the authors.

The collected model data were systematically organized, highlighting the characteristics of the urban and architectural spatial indicators in a clear and structured manner. The spatial indicators were separated according to their themes to serve as provisions for the implementation of urban plans and spatial plans [10,18] (Table 1).

This research investigated urban spatial indicators, including plot size, the proportion of built-up areas, the proportion of landscaped areas, integration with the natural terrain, fencing, road–pedestrian access, and sea moorings (Table 2). Subsequently, architectural spatial indicators were also examined, including the ground floor's floor area, total floor area, height, number of floors, floor plan shape, residential area, auxiliary area, division of the residential section into 1 large and 4 small rooms, and orsans (boathouses) (Table 3).

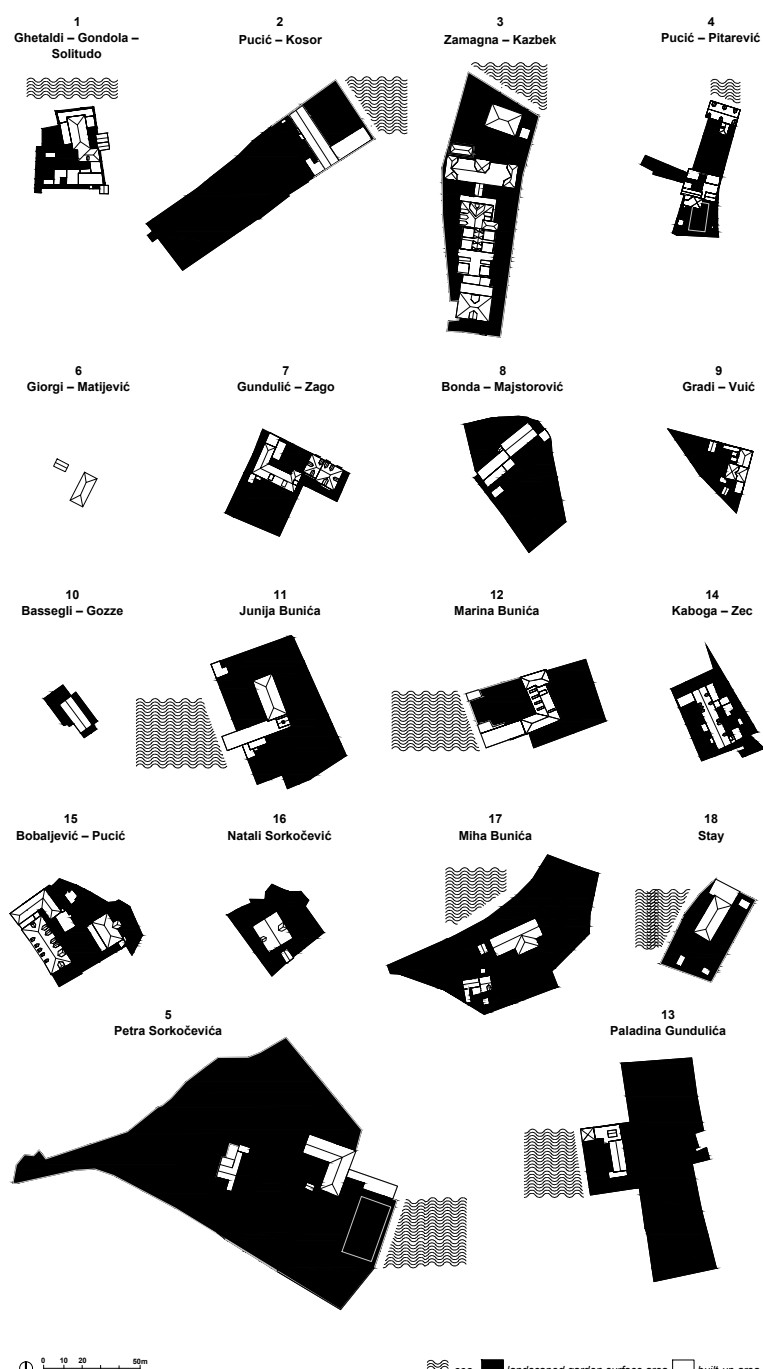

**Figure 4.** Spatial organization of Dubrovnik summer residences. Figure created by the authors.

The average values of the obtained data were determined, and the characteristics of the typology of the summer residences are presented. The average values of the indicators of the historic summer residences formed the basis for expressing the characteristics of the typology of these residences while data were aligned across all types of urban and architectural spatial indicators. Subsequently, an analysis was conducted on the identified spatial indicators for the summer residences and spatial indicators for residential planning in the urban development plans of the city (Figures 5 and 6). The spatial indicators of residential planning were taken from the provisions for the implementation of urban plans for Dubrovnik.

**Table 1.** Indicators of the characteristics of historical Dubrovnik summer residences with modern spatial parameters for planning the development of the City of Dubrovnik, based on the General Urban Plan for the City of Dubrovnik [18].

| Indicator | Goal | Parameter | Unit |
|---|---|---|---|
| Urban and spatial indicators | | | |
| U1: plot surface area | Typology—plot surface area | Plot surface area—average | m$^2$ |
| U2: built-up area | Typology—proportion of the built area in the plot | The share of the built area of the plot—average | % |
| U3: landscaped garden surface area | Typology—proportion of the landscaped garden area | The share of the area of the landscaped garden—average | % |
| U4: integration into the natural terrain | Typology—integration into the natural terrain | The elongated volume of the main building parallels the coast and the sloping terrain—highest value | y/n |
| U5: enclosure | Typology—shape of plot fencing | The presence of a fence wall (>2 m, 2 m, <2 m)—highest value | m |
| U6: vehicular and pedestrian access | Typology—land entrance to the plot | The presence of vehicular and pedestrian access—highest value | y/n |
| U7: mooring area | Typology—sea entrance to the plot | The presence of mooring—highest value | y/n |
| Architectural and spatial indicators | | | |
| A1: ground floor plan area | Typology—ground floor plan area | Ground floor plan area—average | m$^2$ |
| A2: total floor plan area | Typology—total floor plan area | Total floor plan area—average | m$^2$ |
| A3: height | Typology—height | Height—average | m |
| A4: number of stories | Typology—number of stories | Number of stories (ground floor—g, floor—1)—highest value | g,1 |
| A5: floor plan shape | Typology—floor plan shape | Floor plan shape (I, L, U)—highest value | I, L, and U |
| A6: residential area | Typology—proportion of residential area in plot | Share of residential area in the plot—average | % |
| A7: auxiliary area | Typology—proportion of auxiliary area in plot | Share of auxiliary area in the plot—average | % |
| A8: division of the residential section into 1 large and 4 small rooms | Typology—the method of organizing the residential part | The presence of the division of the residential section into 1 large and 4 small rooms—highest value | y/n |
| A9: boathouse | Typology—mooring organization method | The presence of a boat house—highest value | y/n |

**Table 2.** Urban and spatial indicators: U1, plot surface area; U2, built-up area; U3, landscaped garden surface area; U4, integration into the natural terrain configuration; U5, enclosure; U6, vehicular and pedestrian access; and U7, mooring area.

| Dubrovnik Summer Residence | U1 (m$^2$) | U2 (%) | U3 (%) | U4 (y/n) | U5 (m) | U6 (y/n) | U7 (y/n) |
|---|---|---|---|---|---|---|---|
| 1 Ghetaldi–Gondola–Solitudo | 1034 | 65 | 35 | y | >2 | y | y |
| 2 Pucić–Kosor | 4735 | 22 | 78 | y | 2 | y | y |

**Table 2.** *Cont.*

| Dubrovnik Summer Residence | U1 (m²) | U2 (%) | U3 (%) | U4 (y/n) | U5 (m) | U6 (y/n) | U7 (y/n) |
|---|---|---|---|---|---|---|---|
| 3 Zamagna–Kazbek | - | - | - | - | - | - | - |
| 5 Pucić–Pitarević | - | - | - | - | - | - | - |
| 6 Petar Sorkočević | 14,274 | 6 | 94 | y | >2 | y | y |
| 7 Giorgi–Matijević | - | - | - | - | - | - | - |
| 8 Gundulić–Zago | 2011 | 50 | 50 | y | 2 | y | y |
| 9 Bonda–Majstorović | 2791 | 17 | 83 | y | <2 | y | n |
| 10 Gradi–Vuić | 851 | 27 | 73 | y | >2 | y | n |
| 11 Bassegli Gozze | - | - | - | - | - | - | - |
| 12 Junije Bunić | 3841 | 19 | 81 | y | >2 | y | y |
| 13 Marin Bunić | 2252 | 32 | 68 | y | >2 | y | y |
| 14 Paladin Gundulić | 5394 | 7 | 93 | y | >2 | y | y |
| 15 Kaboga–Zec | 1457 | 37 | 63 | y | 2 | y | y |
| 16 Bobaljević–Pucić | 2512 | 42 | 58 | y | 0 | y | y |
| 17 Natali Sorkočević | 1463 | 26 | 65 | y | 0 | y | y |
| 18 Miho Bunić | 1411 | 39 | 61 | y | 2 | y | y |
| 19 Stay | 1915 | 26 | 74 | y | 2 | y | y |
| Average value | 3282 | 29 | 70 | y | >2 | y | y |
| Typology of summer residences | 3500 | 20 | 80 | y | >2 | y | y |

**Table 3.** Architectural and spatial indicators: A1, ground floor plan area; A2, total floor plan area; A3, height; A4, number of stories; A5, floor plan shape; A6, residential area; A7, auxiliary area; A8, division of the residential section into 1 large and 4 small rooms; and A9, boathouse.

| Dubrovnik Summer Residence | A1 (m²) | A2 (m²) | A3 (m) | A4 (g,1) | A5 (I,L,U) | A6 (%) | A7 (%) | A8 (y/n) | A9 (y/n) |
|---|---|---|---|---|---|---|---|---|---|
| 1 Ghetaldi–Gondola–Solitudo | 667 | 992 | 12 | 2 | I | 49 | 51 | y | y |
| 2 Pucić–Kosor | 1060 | 1724 | 12 | 2 | L | 63 | 37 | n | y |
| 3 Zamagna–Kazbek | - | - | - | - | - | - | - | - | - |
| 5 Pucić–Pitarević | - | - | - | - | - | - | - | - | - |
| 6 Petar Sorkočević | 798 | 1058 | 10 | 2 | L | 97 | 13 | n | y |
| 7 Giorgi–Matijević | - | - | - | - | - | - | - | - | - |
| 8 Gundulić–Zago | 1011 | 2022 | 10 | 2 | U | 50 | 50 | n | y |
| 9 Bonda–Majstorović | 475 | 665 | 8 | 2 | I | 40 | 60 | y | n |
| 10 Gradi–Vuić | 229 | 425 | 12 | 2 | L | 86 | 14 | n | n |
| 11 Bassegli Gozze | - | - | - | - | - | - | - | - | - |
| 12 Junije Bunić | 731 | 1012 | 8 | 2 | L | 38 | 62 | y | y |
| 13 Marin Bunić | 723 | 1446 | 8 | 2 | S | 63 | 37 | n | y |
| 14 Paladin Gundulić | 351 | 491 | 10 | 2 | L | 39 | 61 | y | y |

**Table 3.** *Cont.*

| Dubrovnik Summer Residence | A1 (m²) | A2 (m²) | A3 (m) | A4 (g,1) | A5 (I,L,U) | A6 (%) | A7 (%) | A8 (y/n) | A9 (y/n) |
|---|---|---|---|---|---|---|---|---|---|
| 15 Kaboga–Zec | 534 | 839 | 6 | 2 | L | 57 | 43 | y | y |
| 16 Bobaljević–Pucić | 1033 | 2031 | 6 | 2 | L | 25 | 75 | n | y |
| 17 Natali Sorkočević | 374 | 703 | 10 | 2 | I | 88 | 12 | y | y |
| 18 Miho Bunić | 551 | 1102 | 10 | 2 | I | 88 | 12 | n | |
| 19 Stay | 499 | 786 | 10 | 2 | L | 58 | 42 | y | y |
| Average value | 645 | 1092 | 10 | 2 | L | 60 | 40 | y | y |
| Typology of summer residences | 700 | 1120 | 10 | 2 | L | 60 | 40 | y | y |

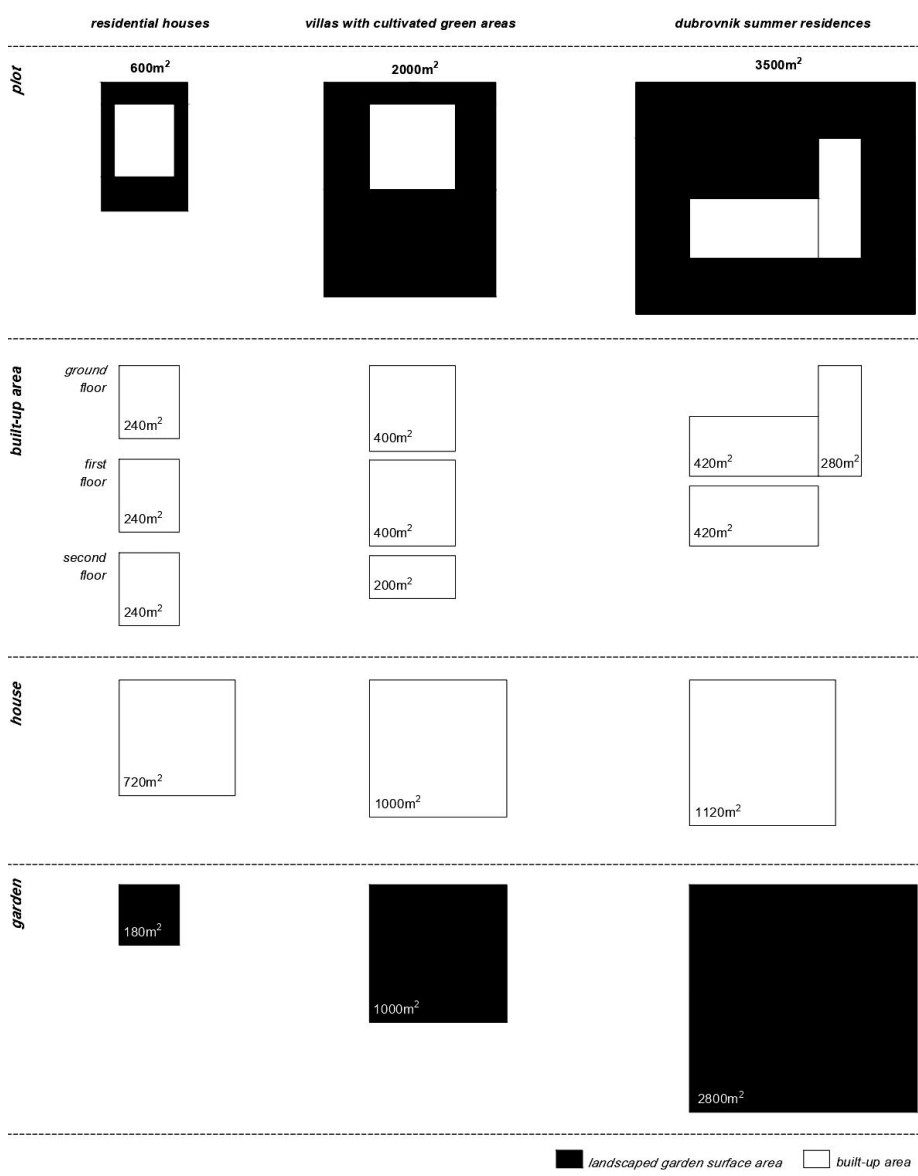

**Figure 5.** Spatial indicators of housing typologies in Dubrovnik. Figure created by the authors.

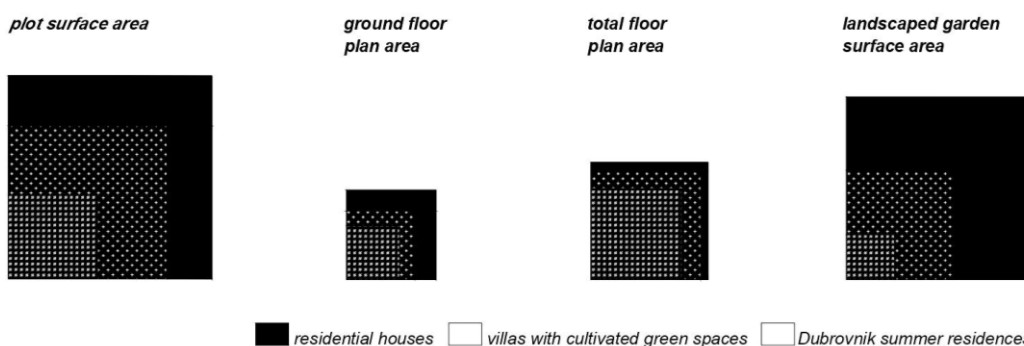

**Figure 6.** Comparison of housing typologies in Dubrovnik. Figure created by the authors.

The obtained results were compared with those of other research, after which, conclusions were drawn regarding the possibilities of using the typology of DSRs as a spatial planning tool for the development of the coexistence between privacy and sociality.

This research delved at a finer scale compared to previous studies, focusing specifically on the dimensions of individual plots and architectural buildings, such as DSRs and the plots belonging to them. However, it is important to note that this research was limited to this narrower scope. Despite this, there is potential to expand this study. Using the indices listed in *Spacematrix: Space, Density and Form* by Meta Berghauser Pont and Per Haupt, a more expansive urban landscape could be systematically explored [22]. These three key indicators are the surface area index (FSI), surface area index (GSI), and network density (N) which are used to draw productive conclusions about urban form and performance. This quantitative approach would enable a comprehensive analysis of the wider urban fabric, evaluating the influence of these parameters on the architectural evolution of the city, instead of exclusively concentrating on a single architectural entity.

In addition, the research could have taken a comprehensive approach to the revitalization of cultural heritage, evolving from an initial focus on individual buildings towards a broader view of the cultural landscape as a connected system of the anthropogenic and natural environments, which was implemented in *The Heritage Urbanism Method* by Mladen Obad Šćitaroci, Bojana Bojanić Obad Šćitaroci, and Ana Mrđa [23]. In this method, identity factors, influencing factors, and value factors are analyzed. Then, the criteria for new interventions, improvement, and revitalization can be defined. However, in this research, we wanted to keep the focus on the urban and architectural features of DSRs and determine which of their specificities can influence the development of the coexistence of privacy and sociality.

## 3. Results and Discussion

The analysis of model data revealed the characteristics of the typology of DSRs, among which some urban and architectural features were prominent, most notably those of the plot, house, and garden, along with the application of technological innovations and the coexistence of privacy and sociality.

### 3.1. Plot

The analysis of the historic DSRs revealed various typological characteristics that unite these houses into a recognizable Dubrovnik rural architectural form. The plot of the summer residence complex (Table 2) has an average size of 3500 m$^2$ (851–14,274 m$^2$), which is significantly larger than the planned residential plots of 600 m$^2$ and the planned villa plots of 2000 m$^2$, which include cultivated green areas. The comparison of the spatial indicators for these plots indicates that there are three different typologies of housing in Dubrovnik (Figures 5 and 6).

The plots show a characteristic arrangement of buildings that form a recognizable L-shaped floor plan with a two-story residential part and a one-story auxiliary part. The majority of the plot around the buildings is landscaped, perfectly integrating the entire

complex into the surroundings to respect the natural topography and utilize the changing terrain to create gardens. The urban and architectural characteristics of DSRs, according to a consideration of all the elements and criteria according to which typology is derived, represent a unique "type that shapes a pattern derived by reducing a group of formal variants to a common basic form" [24].

"However, what makes Dubrovnik's rural architecture interesting, and indeed gives it an exceptional place within a broader framework, is precisely the establishment and persistence of an architectural type that cannot be found elsewhere. In Dubrovnik's rural architecture, spatial "shaping" is a crucial factor in determining the architectural type, which also encompasses the significance of the location where it is built and the spatial-organizational scheme dictated by its function or the totality of functions. Therefore, this architectural type is rightfully referred to as the Dubrovnik type [5]".

"Later, larger country houses were built, first adapted to the requirements of defense, and from the 15th century, when they gradually lost their defensive elements, they grew into luxurious summer houses with an obligatory garden area, in harmony with the surrounding natural landscape. Therefore, the Gruž area was not formed and the emergence of rounded settlements, long-shore lines of a harmonious sequence of built and greened areas were already being created. In the natural tameness of Gruž, with various amenities, zones of rural construction were created, just like in Rijeka Dubrovnik, while in the rest of the territory of the Republic, summer houses appear as individual units, and all this construction, both in terms of quantity and quality, is one of the best chapters in domestic art history [10]". In the past, the typology of DSRs could be characterized as rural. However, as the city expanded and underwent significant urban development, this architectural typology transformed into the urban typology that defines the cityscape today.

The analysis of historic DSRs reveals urban and architectural characteristics that link them into a recognizable form, which stands out from the standard framework of contemporary residential spaces. In comparison to residential or villa plots with cultivated green areas, these residences take on dimensions typical of multifunctional complexes, thereby creating a specific segment of typology that remains unrecognized in the existing urban development plans for the city [25]. Furthermore, it can be determined that plots for the DSR typology did not plan for new complexes in the spatial planning documents for the city's development [18]. Therefore, the question arises regarding the possibility of planning and finding a role for the typology of Dubrovnik's summer residences beyond the framework of residential use. For example, it could potentially be used in some socially significant tasks in the planning and transformation of urban spaces.

*3.2. House*

One of the key features of the summer residences is their large residential area (Table 3). The total average floor area of all the stories is 1120 m$^2$ (ranging from 425 m$^2$ to 2022 m$^2$), which is larger than the standard typology of residential houses with a maximum size of 720 m$^2$, and equally large as villas with cultivated green areas with a maximum size of 1000 m$^2$ (Figures 5 and 6). Similar relationships were identified for the indicators of plot development. Summer residences have an average built-up plot area of 700 m$^2$ (20% of the plot surface area), while contemporary urban plans for residential areas prescribe a maximum built-up area of 240 m$^2$ (40% of the plot surface area), and villas with cultivated green areas have a planned built-up plot area of 400 m$^2$ (20% of the plot surface area). The residential area of the summer house is organized into two units, i.e., the two arms of an "L" shape, in which the average residential part makes up 60% of the surface area (840 m$^2$) over two stories, and the auxiliary part makes up 40% of the surface area (280 m$^2$) on one floor of the building. The standard typology of residential houses and villas with cultivated green areas also allows the total living area to be freely organized in the three stories of the building.

In the 15th century, a characteristic floor plan layout for summer residences began to develop. The house was arranged with one large central room surrounded by four

smaller rooms—"Quattro stanze un salon, ze la casa d'un Schiavon". This configuration reached its peak in the 16th century, thereby defining the spatial characteristics of these residential complexes. The floor plan layout created a dynamic interaction between indoor and outdoor spaces, allowing for the creation of intimate courtyards and terraces while maintaining a balance between privacy and social interactions [26]. The complex consists of two groups of different volumes, forming an L-shaped floor plan. Although the basic "L" shape could be recognized as early as the 15th century, its peak occurred in the 16th century, in terms of both the number of constructed examples and the popularity of the key characteristics. Some of the oldest Renaissance villas near Rome, known as "casali", "casini", or "vigne", which originated in the mid-15th century, exhibit almost the same proportions and the frequent presence of porticos and loggias, similar to the Dubrovnik examples [5].

DSRs from the 15th and 16th centuries have a house located parallel to the seacoast and parallel to the strata of the sloping terrain that usually rises behind the house. On the ground floor, the house has an entrance with a central representative room and peripheral service rooms (Figure 2). From the central room, with a staircase laid to the side, you enter the first floor, where one larger and four smaller rooms are organized. The auxiliary area of the house is perpendicular to the main residential area, and together, they form an "L"-shaped plan. In the auxiliary area, which is laid from the house to the sea, there are service areas and an orsan. On the ground floor, there is often a porch with a terrace in front of the house, which, together with the auxiliary area, frames the representative entrance garden. With the use of the porch on the ground floor, the interaction of the house and the entrance garden is realized, along with the mixing of the functions of the external and internal spaces. On the first floor, all the rooms have direct contact with the upper terrace that surrounds the house. Terraces on the first floor are located as follows: in front of the house, on the flat roof of the porch with open views towards the sea; behind the house, with views and entrances to the back natural or utilitarian garden; and on the side, forming an "L" shape that connects the house to the sea and the back garden.

One can question whether the floor plan layout with "one large room and four smaller ones" should be an integral part of the contemporary forms of typology for DSRs. The exploration of possible floor plan organization forms supported the idea that floor plans are adaptable to contemporary needs. This is also evident in the research conducted on houses designed by influential modern architects [27]. On the other hand, the L-shaped floor plan of a house with a courtyard fits into the common and accepted typological group of houses with courtyards that meet all modern needs and establish an active relationship with the outdoor space [28]. The specificity of the "L" shape of DSRs is reflected in the composition of two wings of the building, consisting of a two-story residential part and a single-story auxiliary part, with the flat roof of the single-story part being used as a raised terrace that extends deep into the garden area of the plot and establishes intimate contact with the natural environment.

*3.3. Garden*

The study of the typology of Dubrovnik's summer residences revealed a dedication to landscaping. The garden, often referred to as a "hortus" or "viridarium", on average, made up (Table 2) almost 80% of the total surface area of the plot (2800 m$^2$), in contrast to the garden of the planned typology of standard residential houses, which amounts to 30% (180 m$^2$). This is also different from the typology of villas with cultivated green areas, where gardens are planned to cover 50% (1000 m$^2$) of the plot area (Figures 5 and 6). The garden areas of all the researched summer residences are enclosed by high walls, providing privacy. This scheme allows for the creation of a completely private courtyard with open terrace spaces for uninterrupted interaction and a connection between the built and garden parts of the complexes.

An extremely important aspect of the development of Dubrovnik's gardens is the continuity passed down from the past. Changes in the design and role of gardens did not

begin in the modern era; their roots are deeply embedded in history. In the Middle Ages, Benedictine orders, especially those based on Lokrum, transmitted experiences from ancient horticulture and applied them in their work. This approach resulted in the development of regular garden beds and innovative plantings that became crucial for shaping garden spaces. "All changes related to the design and role of the garden in rural architecture began long before the 14th century. The bearers of new ideas originated from different places. On the ruins of the Roman Empire, religious orders gathered remnants and experiences of ancient horticulture and transmitted them to various regions where they established monasteries. The Benedictines played a leading role in this endeavor. Their motto 'Ora et Labora' elevates the cultivation of their gardens and estates to more than mere necessity. The Benedictines from the island of Lokrum also arrived and settled in Rožat in 1123, above the Ombla River [5]".

An additional perspective on the role of gardens in summer residences can be seen in villa research, which indicates that gardens were the heart of residences, becoming an extension of the living space and integrating with the architecture itself. Italian gardens and villas further provide insight into different approaches to shaping garden spaces and how they can adapt to the changing demands of modern society [29]. In studies of Mediterranean gardens, it was also observed that they have an influence on society, as they have become places for social interactions, cultural events, and opportunities to showcase the wealth and taste of the owners [30]. Furthermore, research on the connections between residences and urban green spaces in cities has shown that residential structures utilize and integrate green areas into their surroundings. Green spaces in urban contexts contribute to the quality of life of residents and environmental preservation. They are used for recreation, social interaction, and improving the aesthetic aspect of the urban environment [31,32].

Through the analysis of the typology of DSRs, with a focus on studying the relationship between the garden and the house, it becomes clear that gardens are not just aesthetic or utilitarian elements but are equally important as spaces for recreation, relaxation, and social interaction [33]. Gardens are at the heart of DSRs and reflect the way of life in Dubrovnik throughout the centuries. Their role in historical contexts reveals characteristics of adaptability and multifunctionality, indicating an evolutionary potential for meeting the contemporary needs of users and society.

*3.4. Technological Innovations*

The studied complexes of summer residences with stand-alone buildings and an L-shaped floor plan consist of a two-story residential part with a height of up to 12 m and an auxiliary part in a single-story wing; they are characterized by their integration into the terrain of the Mediterranean coastal area through their arrangement of a garden space with unique microclimatic features and enclosed by high walls. These residences can be accessed from the mainland via both vehicular and pedestrian access, as well as from the sea, thanks to a mooring area and boathouse (Tables 2 and 3).

There are elements that indicate that DSRs were extremely technologically advanced. Indeed, the garden, which makes up 80% of the average summer residence's plot, serves as a natural cooling mechanism for the building and an extension of the living space. This creates a microclimate that naturally lowers the temperature during hot summer months, making the garden and summer residence complex more sustainable and environmentally friendly even in modern times [34]. Furthermore, most of these summer residences are integrated into the terrain, following the natural topography and creating terraces that allow for the functionality of the garden and the entire complex.

Researchers studying the evolution of architecture have also noticed technological advancements when comparing 15th-century DSRs with contemporary and traditional villas [5]. Innovative advancements stemming from the contextualization of the natural features of the environment are also evident in research on their cultural, historical, architectural, and construction-related, as well as spatial and ambient, characteristics [35]. Research on summer residential buildings has highlighted numerous possibilities for uti-

lizing their passive characteristics (such as orientation, insulation, ventilation, thermal mass, and shape) to reduce the need for active cooling or heating, harness natural air conditioning, and improve energy efficiency, much like the technological innovations we can observe in the complexes of Dubrovnik's summer residences [36]. In the contemporary context, technological innovations should be carefully harmonized with local cultural and social values in order to ensure their sustainability and authenticity, as has been pointed out by research into sociocultural factors in the transformation of traditional residential architecture [15].

Connecting these experiences with considerations of the contemporary forms of the DSR typology, it can be noted that technological innovations play a significant role in the design and functionality of historic summer residence complexes. Due to their spatial characteristics, they are training grounds for the implementation of innovative breakthroughs, not only for new modern technologies, but also for high energy efficiency, comfort, and sustainability. By combining technological innovations with careful planning and design, the typology of Dubrovnik's summer residences could be envisioned as an intelligent, sustainable, and comfortable space that aligns with the needs of modern life while preserving the traditional architecture and identity of the city.

*3.5. Privacy and Sociality*

It is particularly interesting to note that the relationship between the total floor area of all stories and the garden area (Figures 5 and 6) clearly demonstrates a visible difference between the different typologies: residential houses, villas with cultivated green areas, and DSRs. Residential houses have a total floor area four times larger than the garden area. Villas with cultivated green areas have a total floor area that is equally as large as the garden surface area. The DSR typology has a total floor area three times smaller than the garden area. This peculiarity of summer residences can lead to the discovery of features related to the relationship between privacy and sociality.

Summer residences reflect cultural and architectural heritage, shaping the urban landscape and the community's way of life. They were enclosed by high walls, almost always taller than 2 m (Table 2), which provided a sense of security and isolation while simultaneously offering comfort and enjoyment for social events due to the residences' large plot areas and building complexes surrounded by extensive gardens.

Historically, summer residences were status symbols and signs of belonging to a specific social class. They allowed wealthier individuals to escape the city and crowds during the summer months and served as a place for relaxation, recreation, and social gatherings. In the past, summer residences reflected social stratification and family values. They were places where social interactions, business meetings, entertainment, and cultural events took place [5]. Summer residences, like that of Petar Sorkočević, have long served as a refuge from the hustle and bustle of the city and places for social interactions. Their architectural complex with porticos, terraces, and loggias provided a comfortable stay during the hot summer months. Such summer houses bear a double meaning—private life and public presentation. While they provided privacy within their walls, their presence outside these walls contributed to social displays [37].

Research on the influence of culture on the forms and organization of houses, as well as the impact of form on social aspects and lifestyle, has pointed to the challenges of preserving authenticity in a rapidly changing environment, which can also be observed in the case of DSRs [38]. Contemporary tourism, with its double-edged-sword effect, brings economic benefits but also the risks of commercialization. It makes summer residences exclusive secondary residences, isolated and detached from the community, without a modern social role in the life of the city [13]. Research on the impact of tourism highlights the consequences of the presence of inappropriate forms and intensities of tourist activities, which can lead to the degradation of the urban fabric and the identity of coastal areas [11].

Faced with the challenges of tourism, the typology of Dubrovnik's summer residences, with the planned encouragement of various forms of coexistence of privacy and sociality

(Figure 7), could become a place for community. Shared gardens, social spaces, and events can help build connections among residents, positioning them as essential elements of a vibrant and lively contemporary urban environment [39]. The spatial, functional, and ambient prerequisites for such activities have already been developed in historical summer residences.

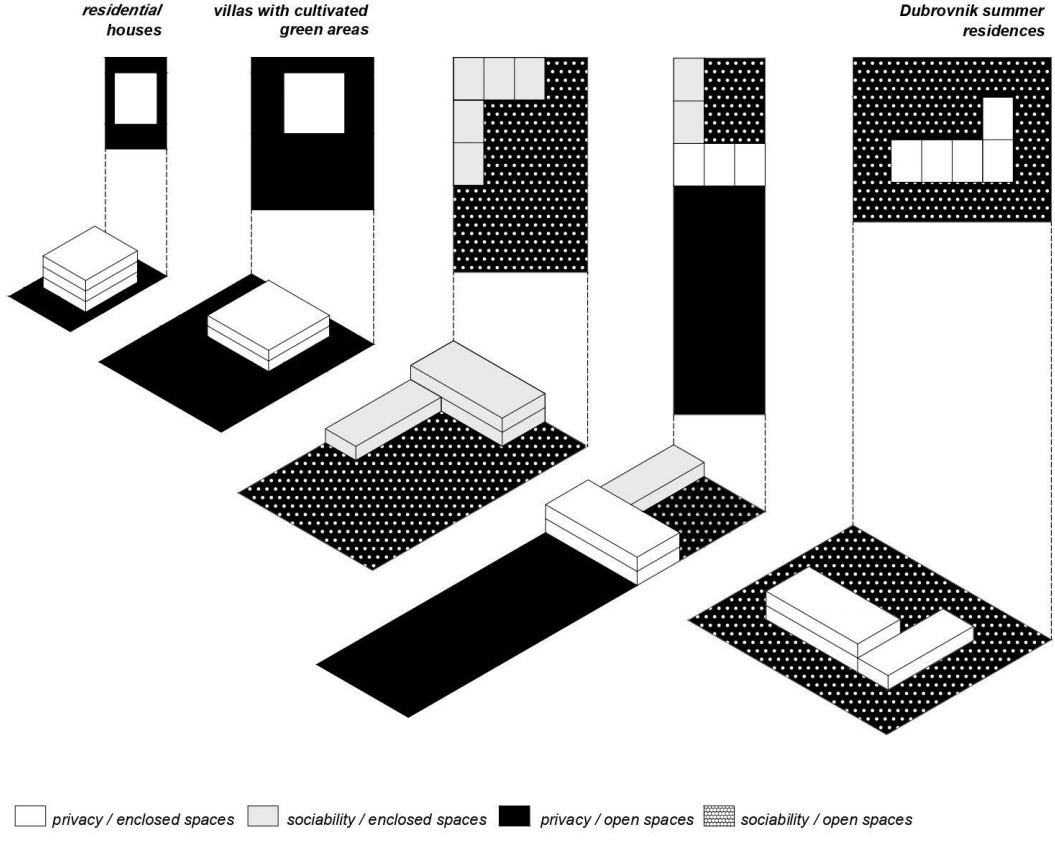

**Figure 7.** Possibilities of using the typology of Dubrovnik summer residences to develop a coexistence of privacy and sociality. Figure created by the authors.

## 4. Conclusions

A model analysis of urban and architectural parameters was conducted on 18 historic summer residences in the coastal area of Gruž. It was based on a model representation with data from cadastral maps and land registers, topographic maps, satellite imagery, urban plans, 3D city models, the literature, and textual data sources, as well as data obtained through field surveys. The urban indicators studied included the plot area, percentage of built-up areas, percentage of landscaped garden areas, integration into the natural terrain, enclosure, vehicular and pedestrian access, and sea access. In addition to those, the following architectural indicators were investigated: the ground floor area, total floor area, height, number of stories, floor plan shape, residential area, auxiliary area, division of the residential area into one large and four small rooms, and boathouses. The limitations of the research results stem from the precision of the numerical and graphical model data.

The research identified specific spatial characteristics of summer residences that have the potential for contemporary use, which concerned the topics of plot, house, garden, technological innovations, and privacy and sociality.

Plot: With a surface area of 3500 m$^2$, DSR plots differ significantly from the plots of residential houses and villas with cultivated green areas. Although the plots of summer residences have been used during the city's development, they have not yet been planned for new developments in the spatial planning documents for future city development.

House: With a surface area of 1120 m$^2$, DSRs have the same house surface area as villas with cultivated green areas, but they have two wings that form an L-shaped floor plan. The two-story residential part makes up 60% of the surface area, while the single-story auxiliary part constitutes 40% of the surface area. The house is organized in such a way that the flat roof of the single-story section is used as a raised terrace that extends deeply into the garden areas of the plot, establishing intimate contact with the natural surroundings.

Garden: With a surface area of 2800 m$^2$, the garden is the essence of a DSR, revealing characteristics of adaptability and multifunctionality, making up almost 80% of the total surface area of the plot, unlike the gardens of residential houses, which constitute 30%, or those of villas with cultivated green areas, which make up 50% of the plot surface area.

Technological innovations: These play a significant role in shaping the design and functionality of DSR complexes. Unlike residential houses and villas with cultivated green areas, these complexes can, due to their spatial characteristics, serve as open arenas for experiments and the implementation of innovative advances and technological progress.

Privacy and sociality: DSRs have a total floor area three times smaller than the surface area of the garden. In contrast, residential houses have a total floor area that is four times larger than the garden surface area, while villas with cultivated green areas have a total floor area equal to the garden surface area. Based on historical experiences of using summer residences, the relationship between enclosed and open spaces, where open spaces are predominantly larger, opens up the possibility for developing different forms of coexistence between privacy and sociality.

The recognized specificities suggest that the typology of DSRs can be used as an exceptionally important urban planning tool for the development of the city's urbanity because it is characterized by a presence in numerous locations along the coast, a large enclosed plot, a large house with two wings for different purposes, and a large garden for various forms of use, enabling the implementation of technological innovations and the development of the coexistence between privacy and socializing.

Examples of the renovation of DSRs show that it is possible to develop the coexistence of privacy and sociality, influence urban development, and encourage social interaction. The Petar Sorkočević summer residence is used by the Croatian Academy of Sciences and Arts for its activities. The summer residence by Miho Bunić is owned by the City of Dubrovnik and used by the Croatian Restoration Institute. Both summer residences have private regimes of use, but at the same time, they are open to visitors, and parts of their closed and open spaces are used for cultural events. The summer residence Sorkočević in Rijeka Dubrovačka, located in the neighboring bay, is owned by the company ACI Marina Komolac and has parts that are for private use. The summer residence is being renovated so that part of the closed interior space can be used as an interpretation center for the presentation of the culture of leisure, and the open exterior space for the presentation of the ambience of the Renaissance Garden. In this way, the summer residence also becomes a place of urban sociality. These examples indicate that the features of summer houses are protected and improved via their social use. Urban transformations, while preserving, restoring, and adapting historical residences, as well as constructing new complexes within the typological framework of DSRs, can create a space that respects private needs while simultaneously encouraging social interactions, the exchange of ideas, and the preservation of the cultural identity of the city and its inhabitants.

The characteristics of DSRs support the fundamental values of the New European Bauhaus—attractiveness, inclusiveness, and sustainability. In addition to realizing the convergence of the relationship "property–society–sustainability", DSRs could also be a medium for implementing the thematic axes of the New European Bauhaus when planning urban development: reconnecting with nature, restoring the sense of belonging and connecting residents, prioritizing places and the people who need them most, and achieving circular sustainability and long-term development goals [1,2].

However, when determining the direction of the future development of the city, it is necessary to include other aspects that affect the space, such as tourism, or tasks

from the management plan, to determine whether there is a possibility of implementing an organizational typology that affirms the advantages of a complex with greater open outdoor spaces than enclosed interior spaces. In addition, other approaches are possible, unique to each DSR, which could result from quantitative and qualitative considerations of the expressed indices that have specific lines of use. The research results can be used in the development of urban and spatial plans, both for conceptualizing their development solutions and shaping their provisions for implementation. Moreover, they contribute to our understanding of the position of the typology of DSRs in relation to contemporary typological forms.

Continuing research can be directed towards verifying the identified features of all parts of Dubrovnik and its urban region where the typology of DSRs appears. Furthermore, expanding this research is possible through a comparison of the characteristics of historical summer residences throughout the entire Mediterranean region while determining their contemporary functions and influence on the development of cities. Research can also be developed in the direction of determining the implementation model for the guidelines of the New European Bauhaus using the typology of DSRs. It is also possible to conduct a more expansive "Spacematrix" study of the urban landscape with the application of indexes of space, surface, and density, which are used to draw productive conclusions about the influence of the urban form of DSRs on the architectural evolution of the city. Likewise, in the continuation of the research of this thematic framework, with the use of the heritage urbanism method, the criteria could be determined for new operations and the improvement and revitalization of DSRs.

Dubrovnik summer residences are an extremely important part of cultural and architectural heritage, and their revival and social reuse are becoming key issues in the contemporary urban environment. This article has been dedicated to the study of the heritage system of DSRs, with a special emphasis on architectural typologies, with the aim of demonstrating their potential for social reuse. The goal was to examine how architectural typologies can serve as valuable spatial planning tools. The first key contribution of this study lies in understanding the value of architectural typology as a spatial planning tool. The analysis of different typologies enables a deeper understanding of the architectural diversity of DSRs and, thus, a better understanding of their spatial relationships and functions. This understanding provides a basis for the development of sustainable urban strategies that ensure the coexistence of cultural heritage and the contemporary needs of communities. The next step in this research is creating a general urban plan and defining a new type of plot. This approach makes it possible to identify the common characteristics of DSRs, which helps in the planning and implementation of revitalization. This study contributes to the revitalization of existing summer residences by providing guidelines for their renovation and rehabilitation according to modern standards. Through a careful analysis of the typology of the building, this study has identified potential methods to maintain the authenticity of DSRs while adapting their functionality to the contemporary needs of the community. Special emphasis was placed on their Mediterranean background and the way the summer residences fit into the environment. The research considered how larger DSR plots, as well as the planning design of plots of the same type, could help protect green areas and incorporate missing public facilities into the urban structure. This approach contributes to the sustainable development of the city, protects the authenticity of the space, and encourages the joint use of resources. The analysis of architectural typologies opens up new perspectives on spatial planning while encouraging the revitalization and preservation of cultural heritage according to the needs of modern society. The examples of renovated DSRs demonstrate the feasibility of integrating historic buildings into contemporary urban life, encouraging social interaction, and preserving cultural identity. This article paves the way for further research, encouraging the development of urban and spatial planning that resonates with the changing needs of contemporary society.

**Author Contributions:** Conceptualization, M.J. (Marijana Jurić), M.J. (Mia Jurić), and K.Š.; methodology, M.J. (Marijana Jurić), M.J. (Mia Jurić), and K.Š.; investigation, M.J. (Marijana Jurić), M.J. (Mia Jurić), and K.Š.; writing—original draft preparation, M.J. (Marijana Jurić), M.J. (Mia Jurić), and K.Š.; visualization, M.J. (Marijana Jurić), M.J. (Mia Jurić), and K.Š. All authors have read and agreed to the published version of the manuscript.

**Funding:** This research received no external funding.

**Institutional Review Board Statement:** Not applicable.

**Informed Consent Statement:** Not applicable.

**Data Availability Statement:** Not applicable.

**Conflicts of Interest:** The authors declare no conflict of interest.

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
