# Peer review of "The Typology of Dubrovnik Summer Residences as a Spatial Planning Tool for Developing the Coexistence of Privacy and Sociality: A Case Study of the Gruž Area"

_heritage, doi:10.3390/heritage6120397_

Round 1
Reviewer 1 Report
Comments and Suggestions for Authors
This is an interesting discussion about the strategic planning of historic summer residences in the Gruz area of Dubrovnik.
The abstract does not make it clear that the focus of the study is specifically about historic buildings, so it is quite a surprise to discover this at the bottom of page 3, section 2. Materials and Methods. The abstract should make this clear. It is difficult to ascertain the exact dates of the buildings and plots analysed, but they may date from the 15th century.
The 18 residences are mapped by spatial parameters, size, size of plot, proximity to the sea, plan shape, etc. The internal organisation is not discussed. The small diagrams (Figure 3) are potentially intriguing, as are the two tables (1 and 2). It would have been good to understand the relationship between the way in which the land and the buildings interacted - for example the view, open terraces and the sun.
There is some contradiction. The introduction describes this as a study of city architecture, but page 9 (line 224) suggests that it is rural. If there is a direct link between the summer residences and the typical Dubrovnik rural typology, then the rural typology must be described.
On page 10, lines 276-282, there is the beginnings of a very interesting discussion about the resemblance of the typical 15th and 16th century Dubrovnik rural typology to the open-plan houses of le Corbusier. Loos's Raumplan is quite different and definitely not the same at all (see Risselada, M. (Ed). ‘Raumplan versus Plan Libre’. Rizzoli 1988)
The conclusions seem to imply that future developments in this area should follow the organisational typology of much greater open outdoor space than enclosed interior space. This seems like a completely different paper. It would require an analysis of tourism in the area. It does not seem that this paper is about that.
This does read a little bit like a first draft, pages 11-15 are individual statements rather than composed paragraphs. The paper maybe needs to be refined.
Comments on the Quality of English Language
The quality of the English is good
Author Response
Response to Reviewer 1 Comments
///
Corrections in the text with responses to the reviewer's comments are marked in red.
///
This is an interesting discussion about the strategic planning of historic summer residences in the Gruz area of Dubrovnik.
The abstract does not make it clear that the focus of the study is specifically about historic buildings, so it is quite a surprise to discover this at the bottom of page 3, section 2. Materials and Methods. The abstract should make this clear. It is difficult to ascertain the exact dates of the buildings and plots analysed, but they may date from the 15th century.
Response
1.1
It is accepted.
Information about the dating of the historical Dubrovnik summer residences was added to the summary.
///
The 18 residences are mapped by spatial parameters, size, size of plot, proximity to the sea, plan shape, etc. The internal organisation is not discussed. The small diagrams (Figure 3) are potentially intriguing, as are the two tables (1 and 2). It would have been good to understand the relationship between the way in which the land and the buildings interacted - for example the view, open terraces and the sun.
Response
1.2
It is accepted.
A section with a description of the summer house was added to the text, with an explanation of the interaction of open and closed space. (pp. 12-13)
In addition, Figure 2 (p. 3) has been added, it shows floor plans and sections of one of the summer houses and explains the relationship between the house and the plot.
///
There is some contradiction. The introduction describes this as a study of city architecture, but page 9 (line 224) suggests that it is rural. If there is a direct link between the summer residences and the typical Dubrovnik rural typology, then the rural typology must be described.
Response
1.3
It is accepted.
The text has been corrected. An explanation for the origin of the typology was added. (p. 10)
///
On page 10, lines 276-282, there is the beginnings of a very interesting discussion about the resemblance of the typical 15th and 16th century Dubrovnik rural typology to the open-plan houses of le Corbusier. Loos's Raumplan is quite different and definitely not the same at all (see Risselada, M. (Ed). ‘Raumplan versus Plan Libre’. Rizzoli 1988)
Response
1.4
It is accepted.
A part of the explanation that creates confusion has been omitted. (p. 13)
///
The conclusions seem to imply that future developments in this area should follow the organisational typology of much greater open outdoor space than enclosed interior space. This seems like a completely different paper. It would require an analysis of tourism in the area. It does not seem that this paper is about that.
Response
1.5
It is accepted.
In the conclusion, an explanation was added emphasizing that in order to determine future development, it is necessary to include other influences in the analysis. (p.17)
///
This does read a little bit like a first draft, pages 11-15 are individual statements rather than composed paragraphs. The paper maybe needs to be refined.
analysis of tourism in the area. It does not seem that this paper is about that.
Response
1.5
It is accepted.
The paper has been refined. (pp. 10-18)
Reviewer 2 Report
Comments and Suggestions for Authors
The study is detailed, clearly exposed and the analyses of the 18 historic summer residences and the definition of common urban and architectural parameters are useful for characterization of the specific typology end to compare to the others, also in Mediterranean Region. But, the possibilities of using the urban typology of Dubrovnik summer residences for urban development planning, was not adequately explained, with some example or hypothesys. How the social interaction could be encouraged, what kind of activity or initiatives could be proposed? In addition, what advantages Dubrovnick could obtain from it, in terms of sustainable approaches?
Comments on the Quality of English LanguageThe reviewer suggests to the authors to improve the fluidity of the paper, limiting repetitions, especially of the subject of some sentences that follow each other (e.g. summer residences). It is suggested to find other substitute grammatical forms.
Author Response
Response to Reviewer 2 Comments
///
Corrections in the text with responses to the reviewer's comments are marked in blue.
///
The study is detailed, clearly exposed and the analyses of the 18 historic summer residences and the definition of common urban and architectural parameters are useful for characterization of the specific typology end to compare to the others, also in Mediterranean Region. But, the possibilities of using the urban typology of Dubrovnik summer residences for urban development planning, was not adequately explained, with some example or hypothesys. How the social interaction could be encouraged, what kind of activity or initiatives could be proposed? In addition, what advantages Dubrovnik could obtain from it, in terms of sustainable approaches?
Response
2.1
It is accepted.
In the paper, a section was added showing examples of the renovation of Dubrovnik summer residences. The examples indicate that the characterization of summer houses is protected and improved by the social use configured in this way. (p.17)
///
The reviewer suggests to the authors to improve the fluidity of the paper, limiting repetitions, especially of the subject of some sentences that follow each other (e.g. summer residences). It is suggested to find other substitute grammatical forms.
Response
2.2
It is accepted.
Fluidity has increased in the overall work. The abbreviation for the term "Dubrovnik summer residence" - DSR is used.
Reviewer 3 Report
Comments and Suggestions for Authors
The article deals with a heritage system, that of the Dubrovnik summer villas, with the aim of justifying its potential for social re-use.
It is stated several times in the text that this approach takes place in a heritage context and that the study aims to justify the fact that building typologies are an exceptionally valuable spatial planning tool.
The reviewer commends the originality of the topic and the systemic view that also leads back to the planning context.
Some notes, however, call for a substantial restructuring and revision of the article.
1. The title speaks of urban typologies, while the article deals with building typologies, there is no analysis or reflection in the text on systems, infrastructures, urban fabric, or anything else, such as to justify the mention of urban typologies in the title.
2. Instead, the text deals extensively with one building typology, that of the summer house in ancient Dubrovnik for the upper-class of Ragusa.
3. The methodology is not up-to-date with recent studies dealing with typological assessment of building and urban sustainability, which is assured by the authors but not demonstrated.
4. Since "Spacematrix: Space, Density and Urban Form" and many other relevant research programs, the indicator parameters of the quantitative elements of building and urban organisms have become increasingly refined and complex. This calls for more detailed motivation in methodology.
5. The text includes quantitative tables of proposed indices and schematic representations. The indices should be accompanied by an adequate bibliography or description that motivates their approriation for evaluation. But it should also be clarified whether it is only the spatial one. Is the cultural assessment of appropriateness entrusted to such indexes?
6. The text challenges the convergence of heritage and reuse considerations in the light of sustainability. However, there are not adequate references and adequate treatment to express heritage recognition and criticize reuse options.
Conclusions
Is the typological characterization of summer houses protected and enhanced by the social use thus configured?
Is it okay for each category or should specific lines of use be associated that follow the quantitative and qualitative considerations made on the expressed indices?
This is just to make some observations and offer some insights. Probably one way would be to submit the study to an urban planning journal. Or, if the intention is a more pertinent convergence of asset, social, and sustainable assessments (as what the New European Bauhaus calls for) we invite substantial integration of the methodology and updating of the structure of the text.
Comments on the Quality of English Language
English does not seem to have been sufficiently reread by a native speaker, in terms of fluency of linguistic structures (still reminiscent of coming from translation) and, in a few cases, approriacy of vocabulary.
Also some ingenious typos are present. For example, In the quotation in inverted commas by Filip de Diversis he can hardly have spoken of a Rijeka Dubrovačka. In the text he may have spoken of 'Ombla Valley' or, reported in English translation, 'Ombla River' or, if updated to the current geographical designations, 'Dubrovačka River' or 'Dubrovačka Creek'.
Author Response
Response to Reviewer 3 Comments
///
Corrections in the text with responses to the reviewer's comments are marked in green.
///
The article deals with a heritage system, that of the Dubrovnik summer villas, with the aim of justifying its potential for social re-use.
It is stated several times in the text that this approach takes place in a heritage context and that the study aims to justify the fact that building typologies are an exceptionally valuable spatial planning tool.
The reviewer commends the originality of the topic and the systemic view that also leads back to the planning context.
Some notes, however, call for a substantial restructuring and revision of the article.
The title speaks of urban typologies, while the article deals with building typologies, there is no analysis or reflection in the text on systems, infrastructures, urban fabric, or anything else, such as to justify the mention of urban typologies in the title.
Instead, the text deals extensively with one building typology, that of the summer house in ancient Dubrovnik for the upper-class of Ragusa.
Response
3.1
It is accepted.
The term "urban typology" is omitted in the title and the entire paper, the term "typology" is used instead. The focus of the work is Dubrovnik summer residences - DSR.
///
The methodology is not up-to-date with recent studies dealing with typological assessment of building and urban sustainability, which is assured by the authors but not demonstrated.
Since "Spacematrix: Space, Density and Urban Form" and many other relevant research programs, the indicator parameters of the quantitative elements of building and urban organisms have become increasingly refined and complex. This calls for more detailed motivation in methodology.
Response
3.2
It is accepted.
Under the title "Materials and Methods" the relationship between recent research and contemporary research is explained. The relationship to the method "Spacematrix: Space, Density and form" [18] and to "The Heritage Urbanism Method" is described [19]. (pp. 9-10)
In the "Conclusion" the possibility of implementing these modern methods is suggested as a possibility to continue the research. (pp. 17-18)
///
The text includes quantitative tables of proposed indices and schematic representations. The indices should be accompanied by an adequate bibliography or description that motivates their appropriation for evaluation. But it should also be clarified whether it is only the spatial one. Is the cultural assessment of appropriateness entrusted to such indexes?
The text challenges the convergence of heritage and reuse considerations in the light of sustainability. However, there are not adequate references and adequate treatment to express heritage recognition and criticize reuse options.
Response
3.3
It is accepted.
An additional table has been implemented in the article (pp. 7-8). Table 1 has the title: Designing indicators of the characteristics of historical Dubrovnik summer residences with modern spatial parameters for planning the development of the city of Dubrovnik, based on General Urban Plan for the City of Dubrovnik [17], created by the author.
The table explains for each indicator: goals, parameters and units.
///
Conclusions
Is the typological characterization of summer houses protected and enhanced by the social use thus configured?
Response
3.4
It is accepted.
In the paper, a section was added showing examples of the renovation of Dubrovnik summer residences. The examples indicate that the characterization of summer houses is protected and improved by the social use configured in this way. (p.17)
///
Is it okay for each category or should specific lines of use be associated that follow the quantitative and qualitative considerations made on the expressed indices?
Response
3.5
It is accepted.
It was added in the paper: "Also, other approaches are possible, unique for each DSR, which could result from quantitative and qualitative considerations of the expressed indices and generate its specific lines of use." (p.17)
///
This is just to make some observations and offer some insights. Probably one way would be to submit the study to an urban planning journal. Or, if the intention is a more pertinent convergence of asset, social, and sustainable assessments (as what the New European Bauhaus calls for) we invite substantial integration of the methodology and updating of the structure of the text.
Response
3.6
It is accepted.
It was added in the paper: "The characteristics of DSR support the fundamental values of the New European Bauhaus - attractiveness, inclusiveness, and sustainability. In addition to realizing the convergence of the relationship "property - society - sustainability", DSR could also be a medium for implementing the thematic axes of the New European Bauhaus when planning urban development: for reconnecting with nature, for restoring the sense of belonging and connecting residents, for prioritizing places and to the people who need it most, and to achieve circular sustainability and long-term development goals [1, 2]." (p.17)
///
English does not seem to have been sufficiently reread by a native speaker, in terms of fluency of linguistic structures (still reminiscent of coming from translation) and, in a few cases, approriacy of vocabulary.
Also some ingenious typos are present. For example, In the quotation in inverted commas by Filip de Diversis he can hardly have spoken of a Rijeka Dubrovačka. In the text he may have spoken of 'Ombla Valley' or, reported in English translation, 'Ombla River' or, if updated to the current geographical designations, 'Dubrovačka River' or 'Dubrovačka Creek'.
Response
3.7
It is accepted.
The text describing Gruž has been corrected. Unclearly written parts have been improved. (p. 2)
Round 2
Reviewer 1 Report
Comments and Suggestions for Authors
The revisions made by the authors have satisfied all of my requests. I recommend that the paper is accepted in its present state.
Author Response
Thank you for your review.
Reviewer 2 Report
Comments and Suggestions for Authors
The paper has been adequately improved.
Author Response
Thank you for your review.